# Lysates of a Probiotic, *Lactobacillus rhamnosus*, Can Improve Skin Barrier Function in a Reconstructed Human Epidermis Model

**DOI:** 10.3390/ijms20174289

**Published:** 2019-09-02

**Authors:** Ye-On Jung, Haengdueng Jeong, Yejin Cho, Eun-Ok Lee, Hye-Won Jang, Jinwook Kim, Ki Taek Nam, Kyung-Min Lim

**Affiliations:** 1College of Pharmacy, Ewha Womans University, Seoul 03760, Korea; 2Severance Biomedical Science Institute, College of Medicine, Yonsei University, Seoul 03722, Korea; 3LCS Biotech, SNU Business Incubator, Suwon 16614, Korea

**Keywords:** probiotic, *Lactobacillus rhamnosus*, skin barrier, skin, reconstructed human epidermis model

## Abstract

The main function of the skin is to protect the body from the external environment. The barrier function of the skin is mainly provided by the stratum corneum, which consists of corneocytes bound with the corneodesmosomes and lamellar lipids. Skin barrier proteins like loricrin and filaggrin also contribute to the skin barrier function. In various skin diseases, skin barrier dysfunction is a common symptom, and skin irritants like detergents or surfactants could also perturb skin barrier function. Many efforts have been made to develop strategies to improve skin barrier function. Here, we investigated whether the microfluidized lysates of *Lactobacillus rhamnosus* (LR), one of the most widely used probiotic species for various health benefits, may improve the skin barrier function in a reconstructed human epidermis, Keraskin™. Application of LR lysate on Keraskin™ increased the expression of tight junction proteins; claudin 1 and occludin as determined by immunofluorescence analysis, and skin barrier proteins; loricrin and filaggrin as determined by immunohistochemistry and immunofluorescence analysis and qPCR. Also, the cytotoxicity of a skin irritant, sodium lauryl sulfate (SLS), was alleviated by the pretreatment of LR lysate. The skin barrier protective effects of LR lysate could be further demonstrated by the attenuation of SLS-enhanced dye-penetration. LR lysate also attenuated the destruction of desmosomes after SLS treatment. Collectively, we demonstrated that LR lysate has protective effects on the skin barrier, which could expand the utility of probiotics to skin-moisturization ingredients.

## 1. Introduction

The skin is the largest organ of the body, which protects our body from the harmful environment and helps to regulate our body temperature. The skin is composed of three layers, epidermis, dermis, and the hypodermis. The outermost layer of the skin, the epidermis, is a multi-layered epithelial tissue, and the barrier function of the skin is mainly provided by the stratum corneum. The stratum corneum serves as a selective permeability barrier against infectious or toxic substances [1]. The stratum corneum is composed of terminally differentiated keratinocytes called corneocytes, that are attached to each other by corneodesmosomes and lamellar lipids. The structural organization of this barrier is also referred as a ‘brick and mortar’ structure with the keratin microfibrils, filaggrin and cornified envelopes forming the bricks and the lipids forming the mortar to seal together the cornified envelopes [2]. Also, tight-, gap-, and adherens junctions formed by desmosomes and proteins like loricrin or filaggrin contribute to the protective skin barrier [3].

Although the barrier is self-maintaining and self-renewing, skin barrier dysfunction commonly occurs in inflammatory skin disorders such as atopic dermatitis and psoriasis [3]. Disruption of the skin barrier causes the collapse of moisture balance in the epidermis, allowing the penetration of allergens or chemicals into the deeper layers of the skin, delaying remission or aggravating the diseases [4,5]. Topical glucocorticoids, a main dermatological pharmacotherapy, can worsen barrier dysfunction and delay recovery [6]. Therefore, the use of skin barrier protective substances to maintain skin hydration and restore barrier function constitutes a major adjunctive therapy for various dermatological diseases [7].

Indeed, recent studies suggest that the complex response of epidermal cells to barrier disruption is crucially involved in the pathogenesis of dermatological diseases. In this context, skin barrier restoration may help the treatment of these dermatological conditions [4]. Topical application of moisturizers formulated with epidermal barrier components like natural moisturization factor (NMF) components [8], amino acids, and ceramides [9,10] could relieve dry skin condition and restore the impaired skin barrier. In addition, emollients, antioxidants, and anti-inflammatory agents [11] are also used for the skin barrier reestablishment [12]. However, still, many efforts are being made to discover novel strategies to recover skin barrier dysfunction.

Probiotics are live microorganisms used for various health benefits on the host. Traditionally, most probiotics are developed as foods or dietary supplements, targeting benefits on the intestine [13]. However, recent studies have shown that probiotics can have favorable effects on heart, liver, lung, and even on mental health [14] as well. There is also escalating interest in the topical application of probiotics for skin health. Bacterial strains that are adherent to the human skin have been studied to identify probiotics [15]. *Lactobacillus rhamnosus* (LR), one of the widely used probiotic strains, enhances intestinal barrier function, and help to prevent intestinal problems. LR exhibits the epithelial barrier protection effect against enterotoxigenic *Escherichia coli* (ETEC) in the porcine intestinal epithelial J2 cells. LR promotes TLR2/Akt activation, which increases tight junction integrity and thus enhancing the barrier function and restricting pathogen invasion [16]. Topically applied LR increased re-epithelization in keratinocyte scratch assays by promoting migration, which is involved in the wound healing pathway [17]. Also, there is a report that *L. rhamnosus* GG increases tight-junction function [18].

However, the effects of topically applied LR on skin barrier function have not been reported, to our knowledge. Here, we investigated whether the microfluidized lysates of LR may improve the skin barrier function in a 3D reconstructed human epidermis, Keraskin™ [19,20]. Following the topical treatment of LR, epidermal structural components of barrier function were investigated using immunohistochemistry, immunofluorescence staining, transmission electron microscopy, and qPCR for epidermal differentiation markers. Protective effects on barrier function were evaluated through measuring cytotoxicity and permeability in the presence or absence of a model irritant, sodium lauryl sulfate.

## 2. Results

### 2.1. Topical Treatment of LR Lysate Increases Epidermal Differentiation Markers of a Reconstructed Human Epidermis, Keraskin™

*Lactobacillus rhamnosus* (LR) lysate was prepared as described in the Methods section and was characterized through examination under a microscope (Figure 1a). LR lysate was topically applied to Keraskin^TM^ every other day for 16 days. Then tissues were stained with H&E (Figure 1b). Sixteen days of culture resulted in excessive generation of the stratum corneum. However, it is evident that LR treated tissues have a more ordered and denser stratum corneum when compared to the control.

To further analyze the effects of LR on the epidermis, the treated tissues underwent immunofluorescence (IF) staining with antibodies against target junction proteins, claudin1, and occludin. IF images show that both tight junction molecules were increased in LR-treated Keraskin^TM^ (Figure 2).

To further analyze the effects of LR on epidermal differentiation, the treated tissues underwent immunohistochemistry with antibodies against cytokeratin 5 (K5), 1 (K1), 10 (K10), loricrin (LOR) and filaggrin (FLG). As shown in Figure 3a and in the scoring of intensity (1 to 3) for each layer of the epidermis (Table 1), LR-treated tissues showed that K5 and K1 advanced into the upper layer of the epidermis; granular layer (GL) and cornified layer (CL), and the intensity increased while control tissues showed K5 and K1 expression limited within the basal layer. In the case of loricrin and filaggrin, there was no significant difference in the localization, but the intensity increased. There was no evident difference in the expression of K10. This pattern was further confirmed with immunofluorescence analysis (Figure 3b).

### 2.2. LR Lysate Increases the Expression of Skin Barrier Proteins, Loricrin and Filaggrin in Keraskin^TM^

To confirm IHC and IF data (Figure 3) and investigate whether LR could promote epidermal differentiation indeed, mRNA levels of loricrin and filaggrin that are major proteins for the formation of the epidermal barrier were evaluated using qPCR. LR lysate was topically applied to Keraskin^TM^ for 2 days and mRNA was isolated as described in the Methods section and qPCR was conducted. As a result, LR-treatment significantly increased the expression of loricrin and filaggrin (by about 60% and 40%, respectively, Figure 4), thus confirming the IHC results.

### 2.3. Topical Treatment of LR Lysate Attenuated Irritant-Induced Cytotoxicity in Keraskin™

Skin barrier protective effects can be examined by determining the protection of irritant-induced cell death [21]. LR lysate was applied to Keraskin™ and an irritant, SLS was treated at various concentrations. As a result, treatment of LR significantly attenuated SLS-induced cell death in Keraskin™ as measured by WST-1 assay (Figure 5a). To visualize the protection effects of LR, Keraskin™ tissues were stained with H&E (Figure 5b). Histologic images also showed that LR treated epidermal tissues had more intact stratum corneum than the control both in the presence and absence of SLS, which was in line with Figure 1b. Especially, while SLS treatment resulted in the almost complete disappearance of stratum corneum, LR treated tissues showed stratum corneum remaining.

### 2.4. Topical Treatment of LR Lysate Reduced the Skin Penetration of Rhodamine B

We further studied the skin barrier protection effect of LR through a skin penetration study. Rhodamine B, a fluorescent dye, was used to evaluate the permeability of the skin. Keraskin™ was treated with LR for 24 h and 0.02% rhodamine B was treated for 2 h. After fixation with OCT compound and cryosection, the permeability of rhodamine B was evaluated with a fluorescence microscope. As shown in Figure 6, fluorescence images confirmed that the permeability of rhodamine B (red) was reduced by treatment of LR both in the presence and absence of SLS.

### 2.5. LR Lysate Protects the Desmosome Degradation in Keraskin™

Desmosomes are intercellular junctions that provide mechanical strength to tissues. They are abundant in epithelial tissues. As desmosomes are one of the essential factors of the skin barrier function, we studied the effect of LR on desmosomes by the transmission electron microscopy (TEM) (Figure 7). When Keraskin™ was disrupted by SLS, the number of desmosomes was decreased in both control and LR treated groups but LR -treated Keraskin™ showed that desmosomes were damaged less by SLS and more preserved than the control groups.

## 3. Discussion

Here, we demonstrated that the lysates of *Lactobacillus rhamnosus* (LR), one of the most widely used probiotic strains for various health benefits, can improve the skin barrier function using a reconstructed human epidermis model, Keraskin™. Application of LR lysate on Keraskin™ increased the expression of target junction proteins, claudin1 and occludin as seen in immunofluorescence images and the expression of skin barrier proteins, loricrin and filaggrin at both protein and mRNA levels. Moreover, the beneficial effects of LR lysate on the skin barrier function could be demonstrated by the attenuation of the SLS-induced cytotoxicity and skin permeability in Keraskin™. Application of LR lysate also attenuated the destruction of desmosomes by SLS.

Keratins form the intermediate filaments which become crosslinked as a cytoskeleton for the cornified envelope, contributing to the stability of the epidermal barrier. The mutations of the keratins cause incomplete intermediate filaments and disturbed barrier function [22]. One of the late epidermal differentiation proteins, filaggrin aggregates the keratin filaments. Many studies demonstrated that the loss of function in filaggrin causes the impaired formation of the epidermis and affect the skin barrier function [23,24]. Loricrin is a major component of the cornified envelope and binds with filaggrin for the formation of the cornified envelope and plays a key role in the skin barrier function [22]. Immunohistochemical images showed that the intensity of cytokeratins 5 and 1 increased and found at the upper layer of the epidermis of LR-treated tissues and the intensity of loricrin and filaggrin also increased, reflecting that LR may promote epidermal differentiation. This was confirmed through qPCR analysis of loricrin and filaggrin. This indicates that the promotion of epidermal differentiation by LR lysate may have increased barrier function through strengthening the stratum corneum, which could be observed with H&E stained tissues.

SLS is an anionic surfactant widely used in various skincare products like soaps and shampoos. SLS causes skin damages through disrupting stratum corneum integrity. We could also observe that the stratum corneum of Keraskin^TM^ was severely damaged by SLS. In addition, it is known that SLS alters the expression pattern of keratinocyte differentiation markers [25]. We demonstrated that LR treatment causes the increased expression of skin barrier proteins at both protein and mRNA levels. So, it is highly likely that LR could enhance the stability and impermeability of the stratum corneum in the presence of SLS though strengthening the stratum corneum, as shown in the histology and permeation study.

Akt activation in keratinocytes promotes epidermal cell differentiation with the increased expression levels of loricrin and filaggrin [26]. Akt activity is also essential for the barrier, especially in the formation of the cornified envelope during the late and terminal differentiation [27]. Akt pathway is generally activated by several extracellular signals such as growth factor, cytokines through phosphoinositide 3-kinase (PI3K)-dependent signaling pathway. Especially, Toll-like receptor 2 (TLR2) is the pattern recognition receptor expressed beneath the stratum corneum and serves to recognize the presence of bacteria. Recently, TLR2 activation resulted in the activation of the PI3K/Akt pathway and has been shown to enhance epithelial barrier function. It is demonstrated that *L. rhamnosus* promotes TLR2/Akt activation in response to bacterial infection, thus protecting cells by maintaining the epithelial barrier and promoting intestinal epithelial cell activation against an enterotoxigenic *Escherichia coli* [16]. In addition, it has been known that bacterial peptidoglycan-induced activation of TLR2 enhances expression of tight junction proteins such as claudins, occludin and ZO proteins thus resulting in enhanced barrier function in keratinocytes [28]. However, it is also reported that LR-augmented tight junction function is not mediated by TLR2 in human keratinocytes [18]. It is thought that topical LR treatment on the skin is derived by different mechanisms from the gut and also application of whole lysates could undergo multiple signaling pathways for barrier protection effects. Currently, further studies are necessary for how LR enhances the skin barrier function.

We examined whether the lysates of LR can have skin barrier protective effects. It would be interesting to examine whether the application of live *L. rhamnosus* can manifest skin barrier protective effects. However, in cosmetics, products must not be contaminated with microorganisms including probiotic species. The FDA regulates that acceptable limits for total (not pathogenic) microorganisms in cosmetics are 500 colony forming units (cfu) per gram in eye-area products and 1000 cfu/g for other area products [29]. So at this moment, the use of live probiotic strains in cosmetic products must overcome this hurdle while other biotherapeutic approaches beyond cosmetics are on-going with live probiotics [30]. If this hurdle could be mitigated by the amendment of cosmetic regulation, new cosmetics for skin moisturization may be possible.

In conclusion, our results support LR has a beneficial effect on the maintaining of skin barrier function. We demonstrated novel skin barrier protection effects of employing a reconstructed human epidermis model. Further studies will be required to elucidate the mechanism of the skin barrier protection effect of LR. With additional researches, LR could be used for the skincare products as the topical probiotics.

## 4. Materials and Methods

### 4.1. Materials and Reagents

*Lactobacillus rhamnosus* (LR) was cultivated in Lactobacilli MRS Broth medium (Difco Franklin Lakes, NJ, USA) at 30 °C, pH 6.2 for 2 days in static conditions. LR culture medium contained protease peptone No. 3 (10.0 g), beef extract (10.0 g), yeast extract (5.0 g), glucose (20.0 g), sorbitan monooleate complex(Tween 80, 1.0 g), ammonium citrate (2.0 g), sodium acetate (5.0 g), K_2_HPO_4_ (2.0 g), MgSO_4_·7H_2_O (0.1 g), MnSO_4_·4H_2_O (0.05 g). LR was harvested and crushed with a microfluidizer (Microfluidics, Newton, MA, USA). Identity of LR was confirmed with gene sequencing for 16s ribosomal RNA to be *Lactobacillus rhamnosus* strain. WST-1 (4-[3-(indophenyl)-2-(4-nitrophenyl)-2H-5-tetrazolio]-1,3-benzene disulfonate) (Roche, Indianapolis, IN, USA), phosphate-buffered saline (PBS), sodium lauryl sulfate (SLS), rhodamine B were purchased from Sigma–Aldrich (St. Louis, MO, USA).

### 4.2. A Reconstructed Human Epidermis Model (Keraskin™)

A reconstructed human epidermis model (Keraskin™) [31] and Keraskin™ culture media were purchased from Biosolution Co., Ltd. (Seoul, Korea). Keraskin™ was placed on a six-well plate filled with 0.9 mL Keraskin™ culture media per well, and pre-incubated for 20–24 h at 37 °C in a humidified atmosphere containing 5% CO_2_. After pre-incubation, tissues were treated with 40 μL of various concentrations of LR diluted in PBS. After 48 h treatment, tissues were rinsed with PBS. LR was treated four or eight times, and control tissues were treated with PBS and followed the same schedule. Culture media was changed every 48 h. After four or eight times of treatment, tissues were treated with various concentrations of SLS diluted in PBS for 10 min and then rinsed with PBS to perturb skin condition.

### 4.3. WST-1 Assay

Tissue viability was determined by WST-1 assay which gives less damage to skin tissues [32]. After incubation, WST-1 diluted in sterile PBS was treated 300 uL for each tissue and incubated for 3 h at 37 °C in a 5% CO_2_ incubator. Then 200 μL of solutions were transferred into a 96-well plate and absorbance was determined by microplate spectrophotometer at 450 nm (BioTek Instruments, Inc., Winooski, VT, USA).

### 4.4. Histological Analysis

For the histological examination, all samples were cut in 10 mm-width and fixed in 4% phosphate-buffered formalin (PFA) for 24 h, as described previously [33]. Fixed samples were paraffin-embedded and cut into 5-μm section using microtome (Leica, RM2235), followed by hematoxylin-eosin staining. Briefly, the sections were stained with 0.1% Mayer’s hematoxylin for 10 min and 0.5% eosin in 95% EtOH. After staining with hematoxylin and eosin, the washing steps were immediately and sequentially proceeded as follows: dip in distilled H_2_O until eosin stops streaking, dip in 50% EtOH for 10 times, dip in 70% EtOH for 10 times, incubated in 95% EtOH for 30 sec and 100% EtOH for 1 min. Then, samples were covered with a mounting solution (Thermo Scientific, 6769007) and examined under the light microscope (OLYMPUS, BX43).

### 4.5. Immunohistochemistry (IHC) and Immunofluorescence Staining (IF)

For immunohistochemistry, skin samples were cut in cut into 5-μm section and were sequentially proceeded rehydration steps with descending graded series of ethanol. Next, pH 6.0 antigen retrieval (DAKO, S1699, Santa Clara, CA, USA) was conducted using a high-pressure cooker for 15 min, followed the cooling phase over 1 h until the solution was fully transparent. After two washes in PBS, sections were incubated in 3% H_2_O_2_ for 30 min for blocking endogenous peroxidase. Another two washes in PBS, sections were incubated with protein block (DAKO, X0909, Santa Clara, CA, USA) for 1–2 h at room temperature in a humidity-controlled chamber. Primary antibodies were incubated overnight at 4 °C. Commercial primary antibodies were purchased and diluted as described below:

Anti-keratin1 (ab185628, 1:1k) anti-keratin5 (ab52635, 1:500), anti-keratin10 (ab76318, 1:3k), anti-loricrin (ab24722, 1:500) anti-filaggrin (NBP1-87528, 1:250). After three washes in PBS, sections were incubated in HRP-labeled anti-rabbit antibody (DAKO, K4003, Santa Clara, CA, USA) for 15 min at room temperature. For the development of HRP-labeled antibody on section, DAB (DAKO, K3468, Santa Clara, CA, USA) was diluted and put it on each section for the identical time. Mayer′s hematoxylin (DAKO, S3309, Santa Clara, CA, USA) was used for counterstaining.

For immunofluorescence staining, skin samples were processed by the same procedure as described in the immunohistochemistry protocol until incubation of primary antibodies. Commercial primary antibodies were purchased and diluted as described below:

Anti-keratin1 (ab185628, 1:1k) anti-keratin5 (ab52635, 1:500), anti-keratin10 (ab76318, 1:3k), anti-loricrin (ab24722, 1:500) anti-filaggrin (NBP1-87528, 1:250), anti-Claudin1 (51-9000, 1:250) anti-occludin (SC-133256, 1:500). After incubation of primary antibodies, samples were immediately washed in PBS two times. Next, samples were incubated with Alexa-568 conjugated anti-rabbit or anti-mouse (Jackson laboratory, Bar Harbor, ME, USA) for 1 h at room temperature. After three washes with PBS, DAPI (Sigma, St. Louis, MO, USA) was used for nuclear staining. All immunofluorescence images were captured by EVOS-FL (Thermo Scientific, Fremont, CA, USA).

### 4.6. RNA Preparation and Quantitative Real-Time PCR (qRT-PCR) for the Determination of mRNA Expression

Total RNA was isolated from Keraskin^TM^ by RNA extraction kit according to manufacturer′s guidelines (Qiagen, Valencia, CA, USA) and a previous report [34]. Then isolated RNA was quantified by spectrophotometer (NanoDrop spectrophotometer; Thermo Scientific, Fremont, CA, USA). Total RNA was reverse transcribed into cDNA using RT master premix with oligo dT (Elpis Biotech Inc., Seoul, Korea) then analyzed by qRT-PCR using an Applied Biosystems 7300 real-time PCR System (Applied Biosystems, Waltham, MA, USA). The sequences of PCR primers were: for loricrin 5′-CCAGACCCAGCAGAAGCAG-3′ (forward) and 5′-TGCCCCTGGAAAACACCTC-3′ (reverse), filaggrin 5′-GAGGGCACTGAAAGGCAAAA-3′ (forward) and 5′-CTTCCGTGCTGAGAGTGTCT-3′ (reverse), β-actin 5′-GCAAAGACCTGTACGCCAAC-3′ (forward) and 5′-GATCTTCATTGTGCTGGGTGC-3′ (reverse). Primers for human loricrin, filaggrin and β-actin were purchased from Cosmo Genetech (Seoul, Korea). Cycling parameters were 50 °C for 2 min, 95 °C for 10 min, 40 cycles of 95 °C for 15 s, and 58 °C for 1 min. For the comparisons, mRNA levels of target genes were normalized to the corresponding β-actin levels in the Keraskin^TM^ and expressed as target gene normalized to β-actin. Relative quantification was performed using the comparative ΔΔCt method according to the manufacturer′s instructions.

### 4.7. Skin Penetration Study

Rhodamine B, a hydrophilic dye, was treated to evaluate the permeability of the skin as described previously [35]. Keraskin™ was treated with LR for 24 h and rinsed with PBS. 0.02% rhodamine B was treated for 2 h and gently washed with an autoclaved cotton swab. Then cultured tissues were initially cut in 10 mm-width and immediately embedded in OCT compound (Sakura, 4583, Tokyo, Japan) using dry ice. Embedded samples were cut into 5-μm width and incubated in room temperature for 40–60 min. Samples were washed in 1X phosphate-buffered saline (PBS) two times for 3 min respectively. After 2 washes, 1000 times diluted DAPI (Sigma, 4583, St. Louis, USA) in 1X PBS were treated on slides which pre-marked by a pen (Enzo, ADI-950-233-0001) for 5 min, followed by three washes in PBS. After then, samples were covered by mount solution (Thermo Scientific, 9990402, Waltham, USA) and immunofluorescence images were captured by EVOS-FL.

### 4.8. Transmission Electron Microscopy (TEM)

Samples were incubated for 24 h in 2% Glutaraldehyde-Paraformaldehyde in 0.1 M PBS and washing in 0.1 M phosphate buffer (PB). Then, samples were post-fixed with 1% OsO4 dissolved in 0.1 M PB for 2 h and dehydrated in an ascending graded series (50–100%) of ethanol and infiltrated with propylene oxide. Specimens were embedded by Poly/Bed 812 kit (Polysciences). After pure fresh resin embedding and polymerization at 65 °C vacuum oven (DOSASKA, TD-700) for 24 h, sections of 200–250 nm in thickness were initially cut and stained with toluidine blue (Sigma, T3260) for light microscopy. 70-nm thin sections were double-stained with 6% uranyl acetate for 20 min and lead citrate for contrast staining. Sections were cut by LEICA EM UC-7 with a diamond knife and transferred onto copper and nickel grids. Desmosomes were examined under transmission electron microscopy (JEM-1011, JEOL) at a voltage of 80 kV.

### 4.9. Statistical Analysis

Results are expressed as the mean ± standard error of the mean (S.E.M) of three or more independent experiments. The statistical analyses were performed by Student’s t-test or two-way ANOVA. *p*-value < 0.05 was considered statistically significant.

## Figures and Tables

**Figure 1 ijms-20-04289-f001:**
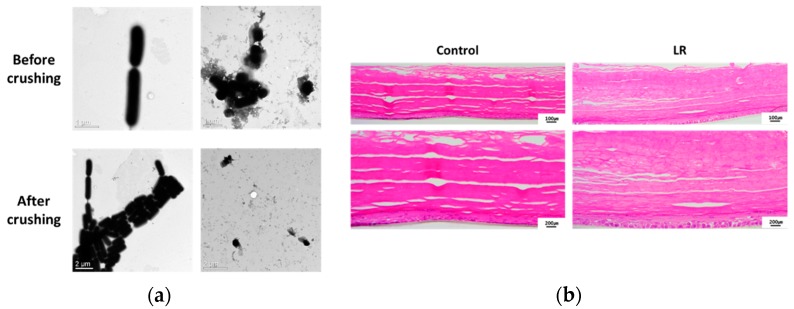
Lysates of *Lactobacillus rhamnosus* (LR) and its effect on Keraskin^TM^. (**a**) Microscopic images of LR lysate before and after crushing. (**b**) Histology of H&E stained Keraskin^TM^ after LR lysate treatment. LR lysate was topically applied to Keraskin^TM^ every other day 8 times. Control; PBS.

**Figure 2 ijms-20-04289-f002:**
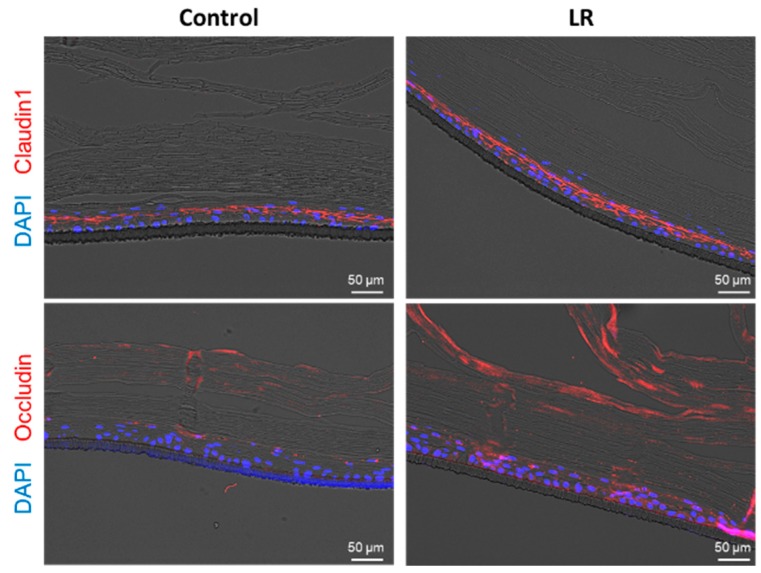
Immunofluorescence images of LR treated Keraskin^TM^ with antibodies against tight junction proteins. Claudin1 (upper red) and occludin (lower red) are stained and DAPI (blue) was used for nuclear staining.

**Figure 3 ijms-20-04289-f003:**
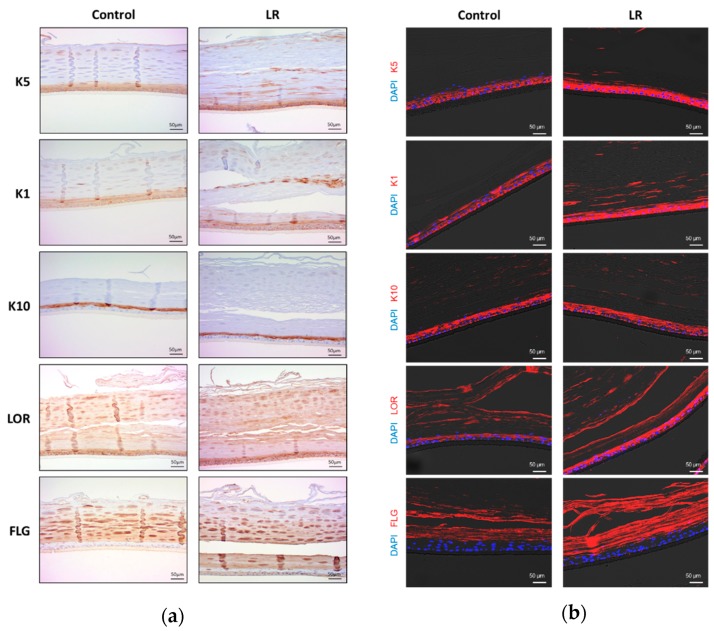
Immunohistochemical and immunofluorescence images of Keraskin^TM^ treated with LR lysate. (**a**) Immunohistochemical images of Keraskin^TM^ with antibodies against cytokeratin 5 (K5), 1 (K1), 10 (K10), loricrin (LOR) and filaggrin (FLG). LR lysate was topically applied to Keraskin^TM^ every other day for 16 days. (**b**) Immunofluorescence images of Keraskin^TM^. DAPI (blue) was used for nuclear staining and red fluorescent indicated lineage markers.

**Figure 4 ijms-20-04289-f004:**
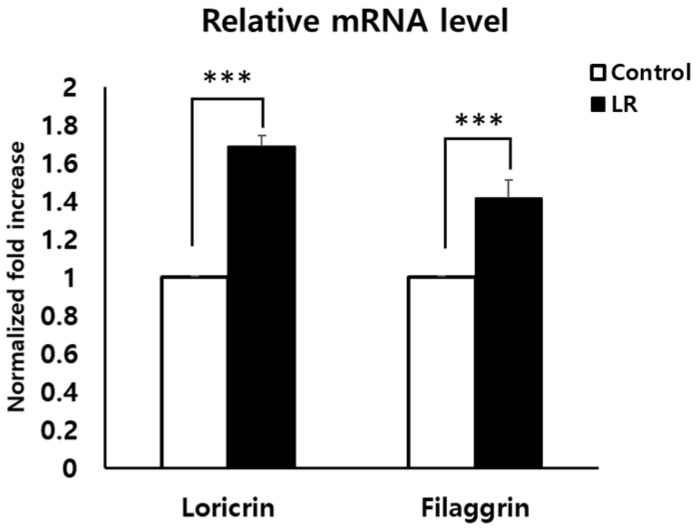
Increases of loricrin and filaggrin expression in LR-treated Keraskin^TM^. mRNA expressions of loricrin and filaggrin were measured by qRT-PCR. LR lysate was topically applied to Keraskin^TM^ for 2 days. Results are expressed as mean ± SEM of independent three tests. *** *p* < 0.001 compared with PBS-treated control.

**Figure 5 ijms-20-04289-f005:**
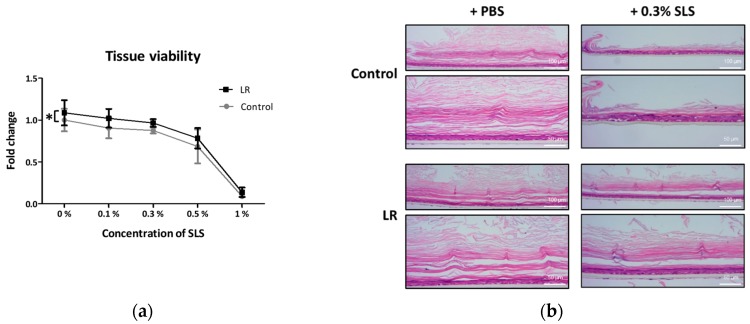
Attenuation of irritant-induced cytotoxicity by LR in Keraskin^TM^. Tissues were pretreated with LR for every other day 4 times and SLS was treated for 10 min. (**a**) Tissue viability was determined by the WST-1 assay. Results are expressed by the fold of control viability as mean ± SE of independent more than three tests and analyzed by two-way ANOVA. * *p* < 0.05 compared with PBS-treated control. (**b**) H&E stained tissues after with PBS or SLS treatment were measured by the light microscopy.

**Figure 6 ijms-20-04289-f006:**
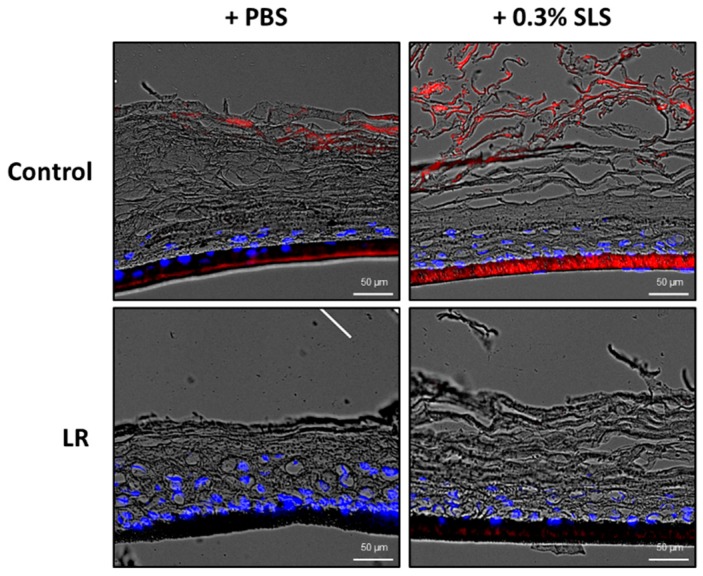
Effects of LR on the skin penetration of rhodamine B. Permeability of rhodamine B was evaluated after 24 h treatment of LR with or without SLS pretreatment. The fluorescent images represented rhodamine B (red), DAPI staining (blue).

**Figure 7 ijms-20-04289-f007:**
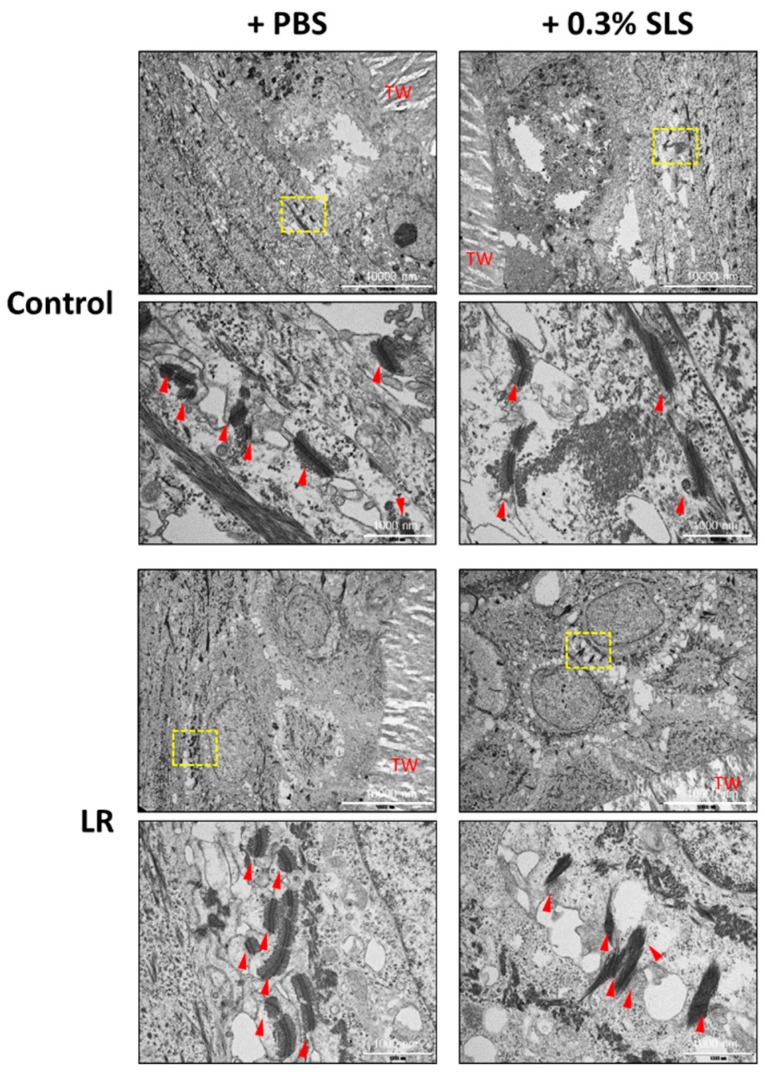
Protective effect of LR on desmosomes in Keraskin^TM^. LR lysate was topically applied to Keraskin^TM^ every other day 4 times and the images of desmosomes in Keraskin^TM^ were measured by TEM. Yellow dotted box indicated the region of higher magnification. Red arrow; desmosome. TW; Transwell

**Table 1 ijms-20-04289-t001:** The localization and intensity score of immunohistochemistry analysis of epidermal differentiation markers.

Target Protein	Test Chemical	Localization	Score (1–3)
K5 (Cytokeratin5)	Control	BL, SL	1
	LR	BL, SL, GL, CL	3
K1 (Cytokeratin1)	Control	BL, SL	2
	LR	BL, SL, GL, CL	3
K10 (Cytokeratin10)	Control	SL, GL	3
	LR	SL, GL	3
LOR (Loricrin)	Control	SL, GL, CL	2
	LR	SL, GL, CL	3
FLG (Filaggrin)	Control	GL, CL	2
	LR	BL, GL, CL	3

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
