# Peer review of "Lysates of a Probiotic, Lactobacillus rhamnosus, Can Improve Skin Barrier Function in a Reconstructed Human Epidermis Model"

_ijms, 2019, doi:10.3390/ijms20174289_

Round 1
Reviewer 1 Report
This manuscript investigates whether lysates of a probiotic, Lactobacillus rhamnosus, can improve skin barrier function in a reconstructed human epidermis model. The research question is clearly worded and the methods are valid. My only question to the authors is why they have used an immunoperoxidase technique and not an immunofluorescence technique. The outcome of the stainings concerns quantitative differences and not qualitative differences. Immunofluorescence is superior when it comes to quantitative differences. Although the expression of K5 and K1, Filaggrin and Loricrin seems to be increased after applying the LS lysates using the immunoperoxidase technique, the differences are small. Immunofluorescence differences would be more reliable. This should be mentioned/discussed in the discussion.
Reviewer 2 Report
Jung and co-authors demonstrated that topical application of bacterial lysate of Lactobacillus rhamnosis (LR) enhanced expression of epidermal barrier proteins and tight junction protein in the organotypic culture of human epidermal keratinocytes. These resulted in increased physical barrier function and protection from chemical barrier damage. This manuscript contains potentially worthwhile information for readers of this journal and in the research areas of skin care and cutaneous biology. However, similar findings have been reported in multiple previous reports. For example, lysate of LR increased Claudin-1, ZO-1 and Occuludin in normal human epidermal keratinocytes, which resulted in enhancement of transepithelial electrical resistance (Appl Env Microbiol 2013, 79, p4887). Similar effects were also observed in wider strains of probiotic strains (International journal of probiotics ad prebiotics, 2012, 7, p81). These data are not discussed in this manuscript. Considering these, the current manuscript lacks originality of the study compared to other articles in this journal and must include further original finding to be publish. Please see specific comments below.
1) It has been widely known that activation of TLR2 enhances expression of tight junction proteins and epidermal barrier function in vivo and in vitro. This must be the most relevant mechanism of action of LR lysate.
2) LR lysate was applied on the 3D skin construct under a sterile condition. This is a serious concern in this manuscript because the mammalian skin is colonized by a vast amount of microbes and the experimental condition tested is not physiologically relevant. Thus, whether or not the topical application of LR brings beneficial effects under a physiological condition is unclear. Therefore, I strongly recommend conducting experiments with mice.
3) Barrier function of 3D skin construct was tested by a rhodamine penetration assay. The data images are not consistent with a typical pattern of this assay. In general, a large quantity of rhodamine is retained on the skin surface and rhodamine penetration is observed as a gradient pattern, with less rhodamine observed on the deeper layers of the skin. However, in the data in this paper, rhodamine is strongly accumulated only on the epidermal layer but not on the surface, especially in the LR-applied skin. The data is not likely due to penetration.
Round 2
Reviewer 1 Report
The authors have adapted the manuscript according to the comments of both reviewers. No further suggestions.
Reviewer 2 Report
The revised manuscript by Jung et al has greatly improved and significantly addressed comments from reviewers.